

# Estimating Seasonal Global Sea Surface Chlorophyll-a with Resource-Efficient Neural Networks

Gabriela Martinez-Balbontin[1, 2], Julien Jouanno[3, 4], Rachid Benshila[3], Julien Lamouroux[1], Coralie Perruche[1], and Stefano Ciavatta[1]

[1]Mercator Océan International, Toulouse, France
[2]Sorbonne Université, Paris, France
[3]Laboratoire d'Etudes en Géophysique et Océanographie Spatiales, Toulouse, France
[4]Institut de Recherche pour le Développement, France

**Correspondence:** Gabriela Martinez-Balbontin (gmartinezbalbontin@mercator-ocean.fr)

**Abstract.** Marine chlorophyll-a is an important indicator of ecosystem health, and accurate forecasting, even at the surface level, can have significant implications for climate studies and resource management. Traditionally, these predictions have relied on computationally intensive numerical models, which require extensive domain expertise and careful parameterization.

We propose a data-driven alternative: a lightweight, resource-efficient neural architecture based on the U-Net that recon-
structs surface, near-global chlorophyll-a from four physical predictors. The model uses mixed layer depth, sea surface temperature, sea surface salinity, and sea surface height as input, all of which are known to influence phytoplankton distribution and nutrient availability. By leveraging publicly available seasonal forecasts of these variables, we can generate six-month chlorophyll-a predictions in a matter of minutes.

We first validated the quality of the reconstruction by using the GLORYS12 reanalysis as input. The reconstructed time series
demonstrated strong agreement with the reference GlobColour observations, with an RMSE of 0.01 and a correlation of 0.95. Extending this approach to seasonal forecasting, we used six-month SEAS5 forecasts as input and found that our predictions maintained high skill globally, with low error rates and stable correlation coefficients throughout the forecast period.

Our model accurately captures spatial and temporal chlorophyll-a patterns across a variety of regions, with an accuracy that meets or exceeds that of the numerical model of reference while significantly reducing computational costs. This approach
offers a scalable, efficient alternative for long-term chlorophyll-a forecasting.

## 1 Introduction

Like many other natural systems, marine biogeochemical cycles are vulnerable to the effects of climate change (Bopp et al., 2013; Achterberg, 2014). Stressors such as rising temperatures (Polovina et al., 2011), ocean acidification (Orr et al., 2005), and changes in nutrient availability (Achterberg, 2014; Schneider et al., 2008) are already affecting the oceans, and these impacts
are expected to intensify in the coming years.

These changes can have cascading effects on marine ecosystems, causing alterations in species distribution and biodiversity, impacting primary production, and posing risks to food security. An important indicator of changes in the health of aquatic



ecosystems is chlorophyll-a (chl-a) concentration. This photosynthetic pigment can be used to estimate phytoplankton biomass and productivity, and it has the additional advantage of being observable from space (Groom et al., 2019), which allows for its
large-scale monitoring.

Accurate seasonal chl-a forecasting, even if limited to the surface level, can have a variety of important applications. It can help estimate food availability for higher trophic levels, with implications for fisheries management and biodiversity conservation (Park et al., 2019b). Additionally, it enables the early detection of harmful algal blooms, allowing for timely warnings to mitigate their impacts on human health and coastal economies (Lin et al., 2021; Wang et al., 2018). Chl-a forecasting also serves
as a valuable indicator of upwelling events, which influence nutrient availability and primary production. These processes, in turn, have effects on the carbon cycle and the broader climate system (Bopp et al., 2013; Achterberg, 2014; Rousseaux et al., 2021; Park et al., 2019b).

Traditionally, biogeochemical forecasting has been done using numerical models, which are based on differential equations that model nutrient and carbon cycles (Aumont et al., 2015; Fennel et al., 2022; Berardi, 2020; Gehlen et al., 2015). While
some of these models have successfully been used for seasonal chl-a forecasting (Rousseaux et al., 2021; Park et al., 2019b), they can be very computationally prohibitive, and they require extensive knowledge of the systems they represent. Incorrect parametrization can lead to systematic biases and uncertainties, and these errors can propagate throughout the model, limiting its capabilities (Fennel et al., 2022; Berardi, 2020; Gehlen et al., 2015).

Machine learning offers the possibility of making use of the growing number of geospatial data for this purpose. While its use
in marine biogeochemical modeling is not as widespread as in meteorology, its adoption is steadily growing (Sadaiappan et al., 2023). In the case of chl-a, it has been used to enhance remote sensing datasets by filling observational gaps and correcting biases caused by sensor limitations or cloud cover (Keiner, 1999; Hu et al., 2021; Kolluru and Tiwari, 2022; Park et al., 2019a; Cao et al., 2020). Martinez et al. (2020) reconstructed historical chl-a patterns from surface oceanic and atmospheric parameters, demonstrating that this approach is able to capture decadal and longer trends of past periods, while Roussillon et al.
(2023) further expanded on this work, highlighting the different regional modes of variability. Machine learning has also been applied to short-term chl-a forecasting, primarily at regional scales. Park et al. (2015) proposed a support vector machine for forecasts within freshwater and estuarine reservoirs, while Wenxiang et al. (2022) used a neural network to generate forecasts in the Xiamen Bay. Additional examples include Chen et al. (2024), who generated forecasts for freshwater lakes, and Ly et al. (2021), who focused on the Han River. Zhu et al. (2023) and Ying et al. (2023) applied data-driven methods for forecasting
in coastal waters. Nonetheless, to our knowledge, most existing efforts focus on regional-scale forecasts spanning only a few days.

Many of these studies demonstrate the feasibility of reconstructing chl-a using physical data (Martinez et al., 2020; Roussillon et al., 2023; Chen et al., 2024; Sauzède et al., 2017). Based on this, we propose a machine learning-driven methodology that can predict global, surface-level chl-a given four oceanic variables: mixed layer depth (MLD), sea surface temperature (SST),
sea surface salinity (SSS) and sea surface height (SSH) at a spatial resolution of approximately 25 km (1/4 degree) and a temporal resolution of 5-days to a month. These variables are known to influence nutrient and oxygen availability, and phytoplankton





distribution, making them good predictors of chl-a (Browning and Moore, 2023; Palacios et al., 2013; Fernandez-Gonzalez et al., 2022; Xu et al., 2022; Barone et al., 2019; Uz et al., 2001; Chenillat et al., 2021).

The goal of this work is to demonstrate that we can not only estimate chl-a from these four variables, but that by using publicly available forecasts of these as input, we are able to generate an ensemble of skillful chl-a predictions for six months into the future.

## 2 Proposed Methodology

### 2.1 Model architecture and training data

The proposed methodology reconstructs a six-month chl-a grid ($\hat{y}$) given an input grid of four physical variables ($X$, i.e. MLD, SST, SSS, SSH). The model is based on the U-Net architecture, which is a convolutional neural network with an encoder-decoder structure that was introduced by Ronneberger et al. (2015). Because of the grid-based, image-like nature of Earth science data, the U-Net and other variations of convolutional neural networks were among the first deep learning architectures applied to tasks such as weather forecasting (Agrawal et al., 2019; Sønderby et al., 2020; Weyn et al., 2019, 2021). While advances in computational power and data availability have enabled the development of more complex architectures, the simplicity and efficiency of the U-Net makes it an effective and resource-efficient choice for this task.

To process both spatial and temporal dimensions simultaneously, we use 3D convolutions (Tran et al., 2015). The encoder consists of two blocks of convolutional and max-pooling layers, which downsample the input. Two additional convolutional layers refine feature extraction before transitioning to the decoder, which mirrors the encoder and upsamples the data back to the original resolution. Skip connections link matching layers in the encoder and decoder, facilitating the transfer of information. A rectified linear unit (ReLU) activation function was used throughout the network, except in the output layer, where a Softplus activation function (Dugas et al., 2000) was empirically found to work best. The model architecture is summarized in Table 1.

The hyperparameters (i.e. number of layers and filters, kernel size, learning rate, etc.) were optimized using random search and the model was trained using the Adam optimizer (Kingma and Ba, 2017). Empirical tests showed that the model had a tendency to underestimate chl-a concentrations, so we modified the standard mean squared error (MSE) loss function by adding a small penalty for underestimation. This adjustment was based on empirical testing and was implemented by applying a weight $w$, where $0.5 > w > 0.3$, whenever the predicted values $p_i$ were smaller than the reference $r_i$. While this approach was effective in reducing underestimation, more principled methods could provide a more robust solution. The physical ocean data was normalized using min-max normalization and the chl-a data was log-transformed.

$$\text{Modified MSE loss function} = \frac{1}{N} \sum [(r_i - p_i)^2 + \max(0, r_i - p_i)^2 \cdot w] \tag{1}$$

To simplify the predictive task, which consists of using a 6-month forecast of the physics to predict chl-a for the same time period, we trained twelve dedicated neural networks, each corresponding to the starting month of the forecast. These networks share a common architecture and fixed parameters. Additionally, the trained weights from any one of the twelve networks can



**Table 1.** Summary of proposed neural architecture

| Layer | Number of trainable parameters | Connected to |
|---|---|---|
| Input layer (input_1) | 0 | - |
| Conv3D (conv3d) | 6,976 | input_1 |
| Conv3D (conv3d_1) | 110,656 | conv3d |
| MaxPooling3D (max_pooling3d) | 0 | conv3d_1 |
| Conv3D (conv3d_2) | 221,312 | max_pooling3d |
| Conv3D (conv3d_3) | 442,496 | conv3d_2 |
| MaxPooling3D (max_pooling3d_1) | 0 | conv3d_3 |
| Conv3D (conv3d_4) | 884,992 | max_pooling3d_1 |
| Conv3D (conv3d_5) | 1,769,728 | conv3d_4 |
| UpSampling3D (up_sampling3d) | 0 | conv3d_5 |
| Concatenation (concatenate) | 0 | up_sampling3d, conv3d_3 |
| Conv3D (conv3d_6) | 1,327,232 | concatenate |
| Conv3D (conv3d_7) | 442,496 | conv3d_6 |
| UpSampling3D (up_sampling3d_1) | 0 | conv3d_7 |
| Concatenation (concatenate_1) | 0 | up_sampling3d_1, conv3d_1 |
| Conv3D (conv3d_8) | 331,840 | concatenate_1 |
| Output (conv3d_9) | 1,729 | conv3d_8 |

be used for transfer learning, where the training process of a model can be initialized with the pre-trained weights of another, optimizing computational cost.

The optimal architecture found for this task has approximately six million trainable parameters. For context, according to some estimates (Shah et al., 2022; Ansari et al., 2022), the original 2D U-Net proposed by Ronneberger et al. has between 20 and 30 million. The training was carried out on a single GPU (NVIDIA A100, 40GB) and took approximately 3-5 hours to complete.

The model was trained using 6-month time series from 1998 to 2016, with physical ocean data from the Global Ocean
Physics Reanalysis (GLORYS12) (Lellouche et al., 2021; Copernicus Marine Service, 2024), and chl-a observations from the Global Ocean Color (GlobColour) dataset (Garnesson et al., 2019). GLORYS12 is a high-resolution reanalysis that provides daily and monthly data on SSS, SST, MLD, and SSH, among other variables. It is based on the NEMO platform and driven at the surface by ECMWF's ERA-Interim and ERA5 reanalyses. Satellite altimetry, SST, sea ice concentration, and in situ temperature and salinity vertical profiles are assimilated using a reduced-order Kalman filter, with a 3D-VAR scheme that
corrects large-scale biases (Lellouche et al., 2021).



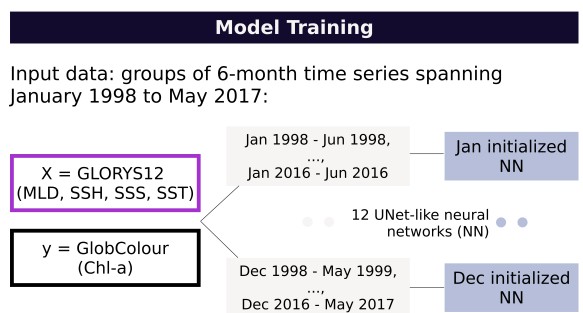

**Figure 1.** A summary of the methodology: Six months of time series spanning January 1998 to May 2017 are used for training twelve neural networks, each initialized on the starting month of the forecast.

GlobColour is a combination of satellite observations of chl-a (alongside other ocean color variables) from multiple remote sensors. Chl-a concentrations are derived from reflectance using sensor-specific algorithm coefficients and then merged. Bias correction is applied directly to the chl-a field, improving consistency and reliability (Garnesson et al., 2019; Maritorena et al., 2010). Since remotely sensed chl-a is limited by sunlight availability, we focus on the -60° to 80° latitudes. Figure 1 shows a brief overview of the methodology.

## 2.2 Validation and prediction generation

We first validated our neural architecture by generating a set of 5-day predictions covering the 2017-2023 period using the reanalysis (GLORYS12) as input. Then, to generate the seasonal predictions, we used 6-month forecasts from SEAS5, the seasonal forecasting system from the ECMWF (Johnson et al., 2019). The forecast consists of a ensemble of 51 members, and it uses the IFS cycle 43r1 for the atmosphere, NEMO for the ocean, and LIM2 for sea ice. The system assimilates in-situ ocean observations (temperature, salinity), satellite altimetry, and snow cover using the OCEAN5 system. Initial conditions for the atmosphere and land surface come from ECMWF operational analyses, and forcings include CMIP5 historical and RCP 3-PD greenhouse gases (Johnson et al., 2019).

The SEAS5 forecasts are available free of charge on the Copernicus Climate Data Store at a monthly, one-degree resolution (Copernicus Climate Change Service, 2018a, b). A high resolution (1/4 degree) version of the data was kindly provided by the ECMWF, but it is possible to obtain similar results by downscaling with a simple interpolation method.

The SEAS5 forecast has some minor differences with respect to the GLORYS12 reanalysis. To avoid these from biasing our evaluation of the model, we minimized them by calculating the difference between the forecast and reanalysis climatologies, and then subtracting this difference from SEAS5 before using it as input.

We used the first 10 (out of 51) ensemble members from SEAS5 as input for the neural network, and generated six-month predictions spanning 01/2020-05/2024. We then concatenated the $nth$ month of each 6-month chl-a prediction to obtain five




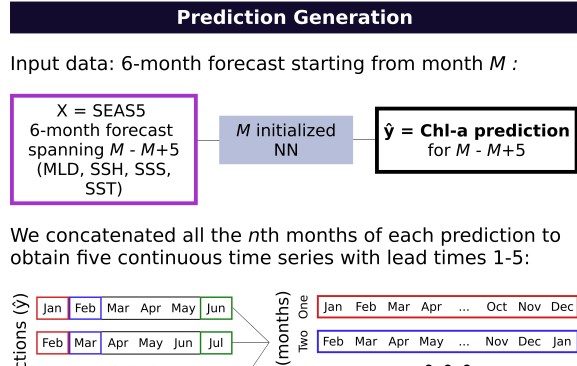

**Figure 2.** A summary of the prediction generation workflow: We concatenated the $nth$ months from each prediction to generate continuous time series with lead-times 1-5.

continuous time series: 01/2020-12/2023 for one-month lead-time, 02/2020-01/2024 for lead-time two, etc. This is illustrated on Figure 2 for clarity.

We evaluated performance by calculating the Anomaly Correlation Coefficient (ACC) for each longitude-latitude grid point as follows :

$$\text{ACC} = \frac{\sum (r_i - c_r)(p_i - c_p)}{\sqrt{\sum (r_i - c_r)^2 \sum (p_i - c_p)^2}} \qquad (2)$$

Where $r_i$ and $p_i$ are the reference (i.e. GlobColour) and predicted values, respectively, and $c_r$ and $c_p$ are the climatologies of the reference and predicted values, respectively.

We also report Root Mean Squared Error (RMSE), the Pearson correlation coefficient (r), normalized RMSE (NRMSE) and
normalized Mean Absolute Error (NMAE). The latter two are calculated with respect to the standard deviation of the reference data:

$$\text{NRMSE} = \frac{\sqrt{\frac{1}{N} \sum (r_i - p_i)^2}}{\sigma_r} \qquad (3)$$

$$\text{NMAE} = \frac{1}{N} \sum \frac{|r_i - p_i|}{\sigma_r} \qquad (4)$$

We analyzed performance by zooming into some regions of interest (illustrated on Figure 3): the North Pacific (170°E-
130°W, 25°N-45°N), Tropical Pacific (170°E-100°W, 10°S-10°N), South Pacific (170°W-100°W, 35°S-15°S), North Atlantic



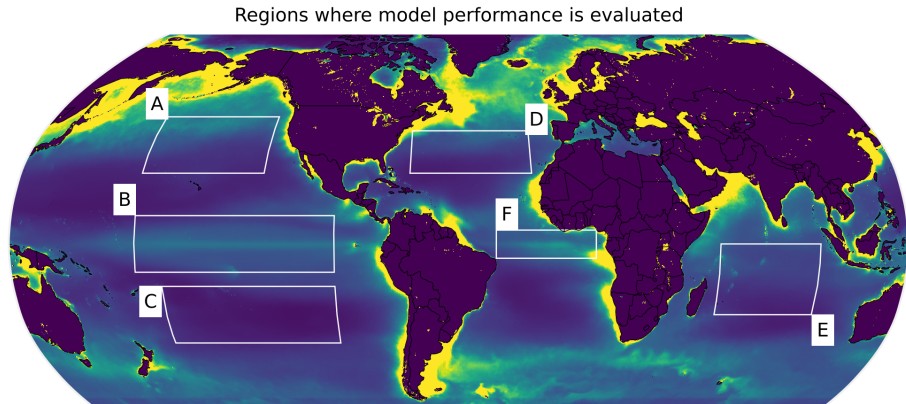

**Figure 3.** Regions where performance is evaluated: North Pacific (A), Tropical Pacific (B), South Pacific (C), North Atlantic (D), Indian Ocean (E), and Equatorial Atlantic (F).

(70°W-20°W, 25°N-40°N), and Indian Ocean (55°E-95°E,25°S-0°S), which are based on Park et al. (2019b), as well as the Equatorial Atlantic (35°W-5°E, 5°S-5°N).

We evaluated our results with respect to the state-of-the-art global biogeochemical analysis BIO4 from Mercator Océan International (Copernicus Marine Service, 2023), which is based on the PISCES model, the biogeochemical component of the NEMO platform (Lamouroux et al., 2023; Aumont et al., 2015). BIO4 models the cycles of carbon, oxygen, key nutrients, and two functional groups of both phytoplankton and zooplankton, generating a 3D global-scale representation of 24 biogeochemical variables. It provides both an analysis and a 10-day forecast at depths of up to 5700m, governed by internally consistent differential equations that capture the dynamics of ecosystems globally. BIO4 is forced by GLO12, a near real-time system and forecast based on the same physics reanalysis (GLORYS12) used for training in our study. Additionally, BIO4 assimilates chl-a observations from GlobColour on a weekly basis (Lamouroux et al., 2024).

Since our focus is on surface chl-a, we evaluated our results against the surface output of the BIO4 analysis. Rather than a direct comparison, we use BIO4 as a benchmark, recognizing that it simulates a wide range of interconnected biogeochemical processes across various depths, whereas our data-driven approach is specifically designed for surface chl-a prediction.

## 3 Results

To validate the neural architecture, we first generated a set of 5-day predictions covering the 2017-2023 period using the reanalysis (GLORYS12) as input, illustrated in Figure 4. The spatially-averaged reconstructed time series has a RMSE of 0.01, NRMSE of 0.32, NMAE of 0.24, and a correlation coefficient of r=0.95.

We then generated a 10-member ensemble of seasonal predictions using the SEAS5 forecasts as input. These six-month predictions span from Jan-Jun of 2020-2023 to Dec-May of 2020-2024 (see Fig. 2 for details) at a 1/4 degree, monthly resolution. On Figure 5 we can see the ensemble of predictions initialized on January, April and June, with these months highlighted.





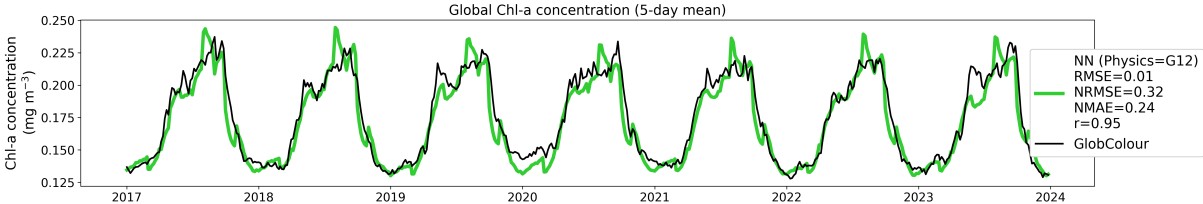

**Figure 4.** Chlorophyll-a concentration: Neural network (NN)-generated prediction with GLORYS12 (GL12) as input physics (green), and the reference observations (GlobColour) in black.

These months correspond to lead-times one out of the six months of each forecast. While the predicted values tend to follow the observations closely, the model is not without its limitations. Figure 5 shows that while the SEAS5-driven neural network accurately captures the start and end of the bloom's global average, it tends to underestimate its peak. In contrast, the GLORYS12-driven predictions (Fig. 4) slightly overestimate the peak of the seasonal cycle, and shift the timing of the maxima 160 in almost every year.

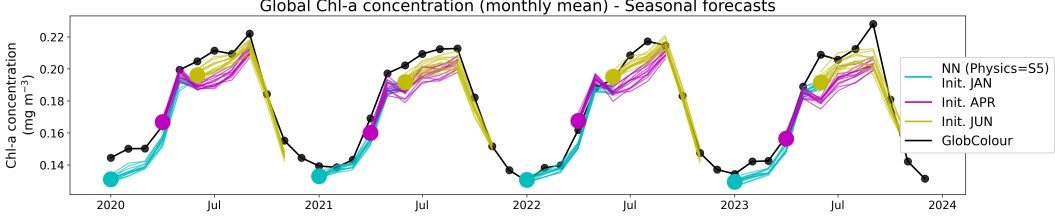

**Figure 5.** Seasonal chl-a forecasts (10 members) initialized on January, April and June generated using SEAS5 (S5) as input physics (shown cyan, magenta and yellow, respectively). The circular marks correspond to the lead-time one of each forecast.

We concatenated all the $nth$-lead-time predictions to generate continuous time series (see Fig. 2). Figure 6 shows the Anomaly Correlation Coefficient (ACC) for lead-times one and six months. The ACC ranges from 1 (indicating perfect correlation) to -1 (complete anticorrelation) with 0 signifying no correlation. The figure shows that the neural network-generated predictions have relatively high skill globally, with ACC values above 0.6 across various regions at lead-time one. As expected, 165 the ACC decreases with lead time due to the increased uncertainty in the forecasted physics. However, despite the deterioration, positive correlations persist in numerous areas.

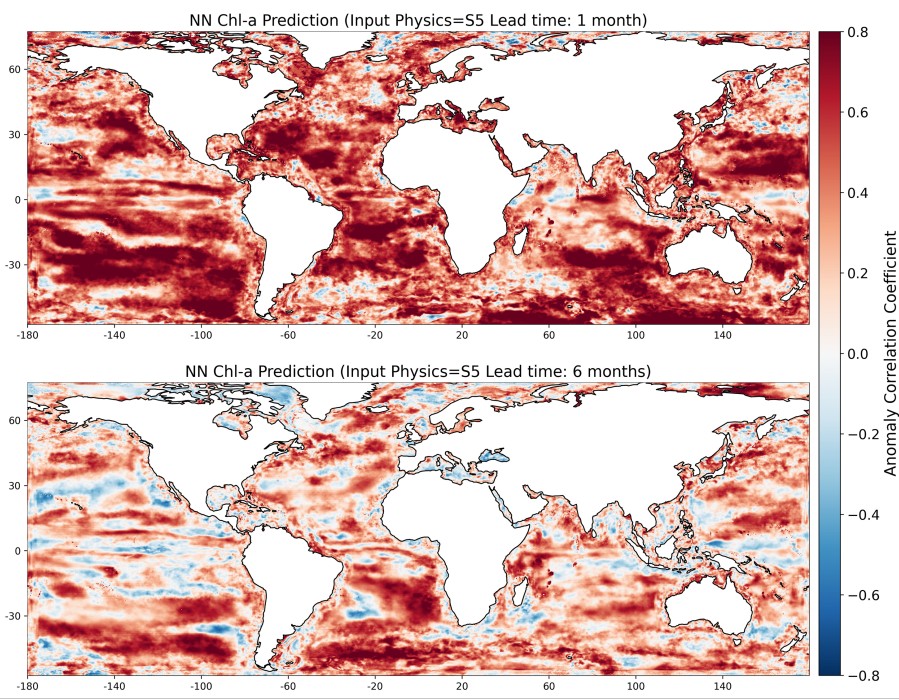

**Figure 6.** Anomaly correlation coefficient (ACC) for the generated predictions when using the SEAS5 (S5) forecast at lead-times one and six (months). To better highlight predictive skill, the color scale limits were set to -0.8 and 0.8.



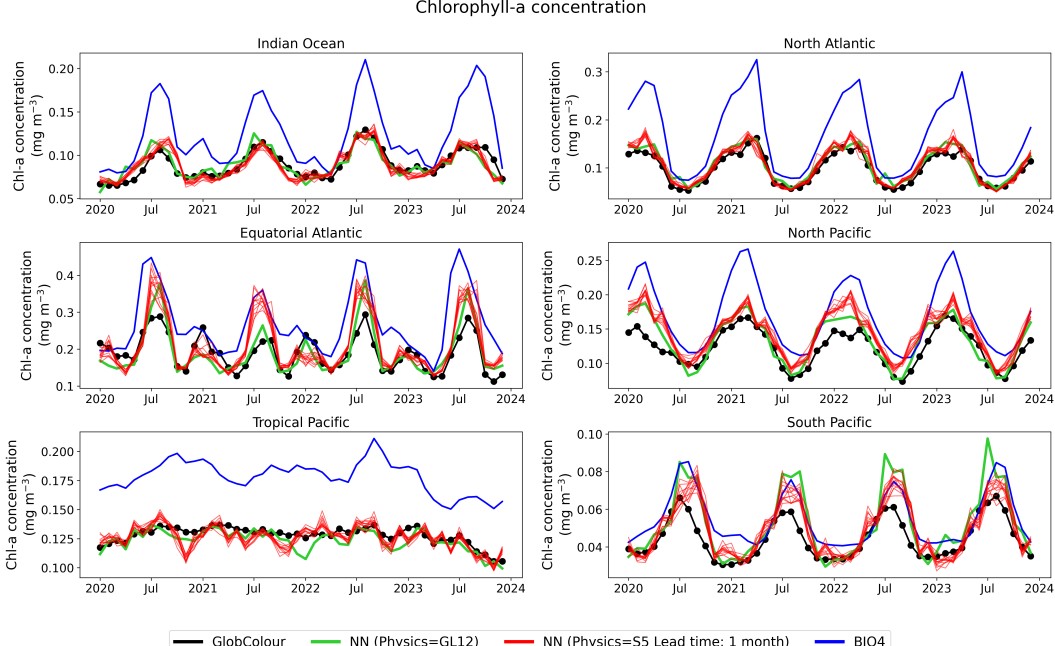

**Figure 7.** Monthly surface-level chl-a time series for each region. The reference dataset (GlobColour) is shown in black, the ensemble of the NN-generated predictions based on SEAS5 (lead-time one) in red, the prediction based on GLORYS12 in green, and the BIO4 output in blue.

Figure 7 shows the lead-time one predictions based on SEAS5 as input, those based on GLORYS12, the GlobColour observations, and the corresponding output from BIO4 for the regions illustrated in Fig. 3. Figure 8 shows the Hovmöller diagrams for each region. These figures demonstrate that the neural network is able to capture the seasonal dynamics of the data across regions, regardless of the input physics.

In certain regions from Fig. 7, the transitions of the neural network's predictions can appear more abrupt than those of the observations and the numerical model. This is particularly noticeable in the Equatorial Atlantic, and the Tropical and South Pacific, where the time series have seemingly sharper increases and decreases in concentration. This abruptness may be partially due to the way in which the time series is constructed, as the predictions are based on a concatenation of all the lead-one months (see Fig. 2). Since each prediction may contain its own biases, the process of stitching them together can introduce discontinuities, leading to these sudden shifts in concentration values. The magnitude of these errors can be further examined in Fig. 8, which shows the chl-a values along the time (x-axis) and latitude (y-axis), and allows for a more detailed assessment of the evolution of the predictions spatially and temporally.



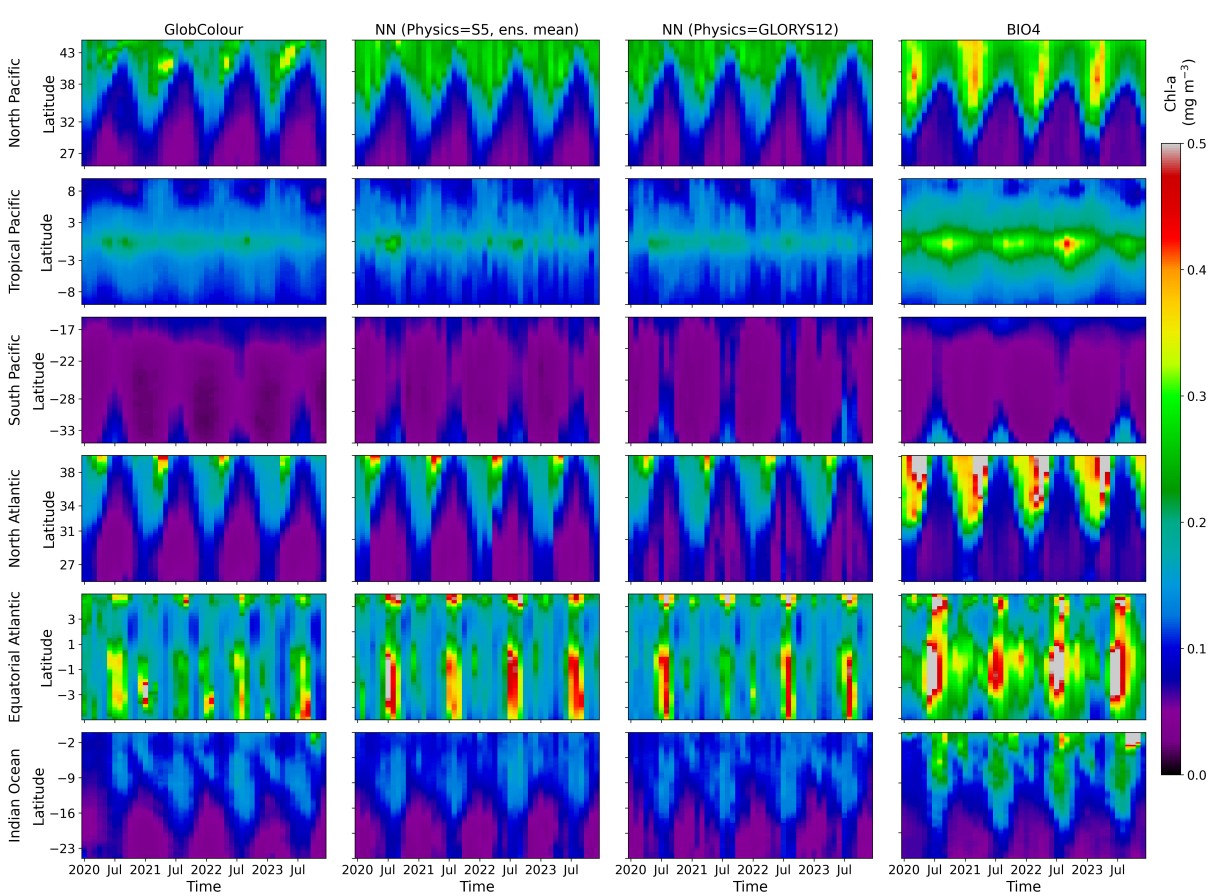

**Figure 8.** Hovmöller diagrams for each region: the first column shows the GlobColour reference dataset, the second shows the ensemble mean of the lead-time one SEAS5-based prediction, the third shows the GLORYS12-based prediction, and the fourth shows the BIO4 output.





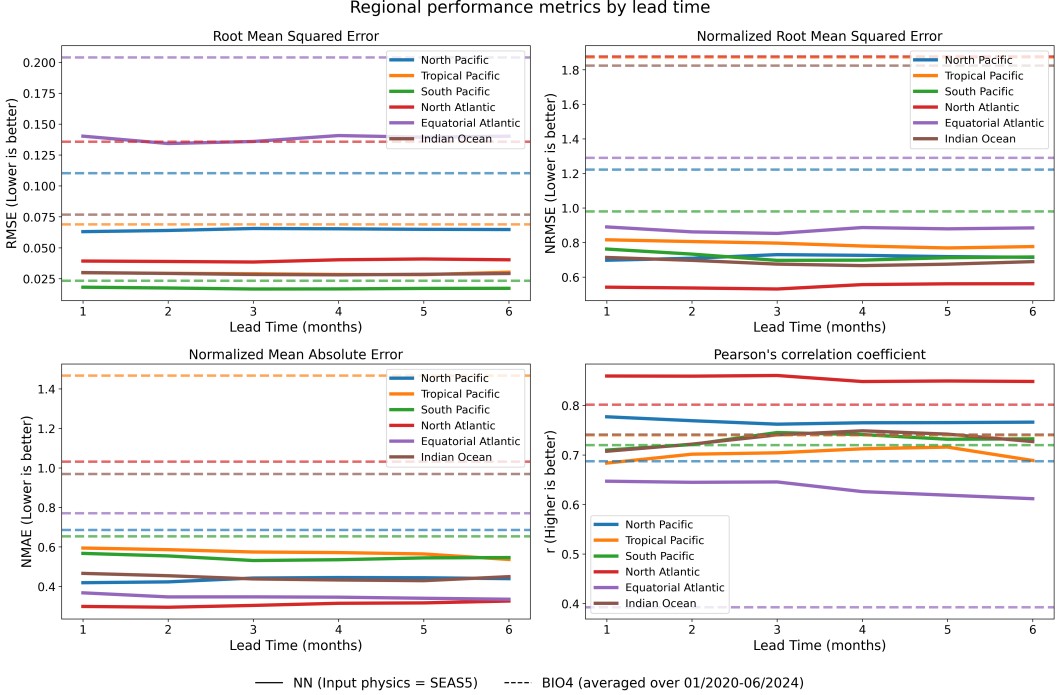

**Figure 9.** Metrics for different regions: RMSE, NRMSE, NMAE, and correlation (r), across different lead-times.

Figure 9 illustrates the evolution of RMSE, NRMSE, NMAE, and the correlation coefficient (r) with lead-time for each
region. We also show BIO4's performance (dotted lines) averaged over the period of study (01/2020-05/2024) for reference.
The figure shows that the neural network tends to have lower error than the corresponding BIO4 output in most regions. Additionally, the neural network's error rates and correlation coefficients remain relatively stable throughout the forecast period.
This sustained performance with lead-time might seem contradictory to the decrease in ACC observed in Fig. 6, but it is likely
that the neural network's ability to capture the strong seasonal dynamics in the data (Figs. 7 and 8) is compensating for the
decrease in performance with respect to the anomalies. The decrease in ACC with increasing lead-time suggests that the neural
network is less effective at predicting the exact magnitude and location of the anomalies over longer periods. However, Fig. 9
indicates that the model might still be good at capturing the general trends and overall patterns within each region, even as the
forecast horizon increases.





## 4    Discussion

We introduced a simple neural network architecture that is capable of reconstructing surface, near-global chl-a from only four physical variables (MLD, SSH, SSS and SST) with relatively high accuracy and low error. A key advantage of our approach is its computational efficiency: the model can be trained in just few a hours on a single GPU, and it generates 1/4-degree, 6-month predictions within minutes. Our method maintains performance regardless of the source of the input physics data, and its accuracy can be further enhanced by correcting the inputs with a GLORYS12 climatology before use.

Our approach aligns with previous studies that have demonstrated the potential of using physical ocean variables and machine learning for chl-a reconstruction (Sauzède et al., 2017; Martinez et al., 2020; Roussillon et al., 2023; Chen et al., 2024). By applying it to forecasted physical ocean states, we further expand the utility of this approach to forecasting, offering a simple yet useful framework for seasonal, near-global surface chl-a predictions.

We evaluated forecasting skill through the Anomaly Correlation Coefficient (ACC), error rates (RMSE, NRMSE, NMAE)
and Pearson's correlation coefficient (r). While some degradation in ACC is observed with increasing lead-time (Fig. 6), correlation and error rates on the raw time series remain satisfactory (Fig. 9). We argue that this discrepancy may be explained by the neural network's ability to capture the strong seasonal patterns of the data (see Figs. 7 and 8), potentially compensating for the increased uncertainty in anomaly prediction over longer forecasting horizons.

### 4.1    Limitations

Despite its promising results, our approach has several limitations. The decline in ACC with increasing lead-time suggests that while the model might effectively capture general seasonal patterns, it may struggle in more complex scenarios where anomalies and outliers are crucial. This could also indicate that the neural network may be overfitting to certain historical patterns.

When comparing our model to the BIO4 analysis, we find that BIO4 exhibits a more pronounced bias towards overestimation,
a tendency that is also present in other traditional biogeochemical models (Rousseaux et al., 2021). Data-driven methods like ours, which learn directly from observations, might be better equipped to capture the actual pixel-value magnitudes of the data by avoiding the parameterizations that could be giving rise to these biases. However, the complexity of numerical models such as BIO4 also comes with key advantages. The numerical model accounts for a wide range of interconnected biogeochemical processes and it ensures, by design, internal consistency across all simulated variables. As a result, it may be more reliable
for extrapolating beyond previously observed conditions, whereas data-driven approaches can struggle with out-of-sample predictions if the training data lacks sufficient variability or if the model is not sufficiently robust.

Our shallow neural architecture and the decision to limit the input to four variables were intended to create a lightweight and efficient framework. However, this comes at the cost of potentially failing to capture the full intricacy of the underlying processes, and some applications may require a more sophisticated approach for greater robustness. This challenge of general-
ization can be further influenced by the use of the MSE loss function in the training process. While MSE is a standard choice for regressive neural networks such as this one, its focus on minimizing error can lead to overly smooth predictions that tend





toward the mean. This can result in the loss of fine-scale detail in regions with high variability, and an overall impact on the realism of the model's predictions.

Finally, we present an "ensemble" of predictions that is derived from the reconstruction of chl-a from the input forecast ensemble. By design, this approach inherits the uncertainties and biases of the SEAS5 forecast, without accounting for the full range of uncertainty linked to the prediction of chl-a. A more robust ensemble generation approach would require the inclusion of more comprehensive sources of uncertainty and the perturbation of these within the predictive framework.

### 4.2 Future directions and improvements

The performance of our model is inherently linked to the quality and diversity of the training data. Research suggests that chl-a has long-range predictability (Rousseaux et al., 2021; Park et al., 2019b; Séférian et al., 2014; Ham et al., 2021; Frölicher et al., 2020), with interannual skill closely linked to biogeochemical drivers, which may exhibit longer predictability than physical variables such as SST (Séférian et al., 2014; Rousseaux et al., 2021; Park et al., 2019b). Similar to nutrient anomalies, SSS anomalies are influenced by MLD variability, and both of these have been identified as important factors in predictability (Fransner et al., 2020; Séférian et al., 2014; Park et al., 2019b).

This suggests that while our model may be capturing some of these key physical processes, it could be further improved by integrating biogeochemical variables as predictors. Expanding the training dataset to include a broader range of physical and biogeochemical drivers, as well as incorporating longer time series, vertical profile data, and more diverse sources of information, would almost certainly improve the model's predictive skill.

Another pathway for improvement could be scaling the neural architecture itself. This could be achieved by adding more of the same components (i.e. more convolutional layers, a deeper encoder-decoder structure, etc.), or by integrating it with other types of architectures. For example, diffusion-based approaches and probabilistic methods could enhance the ensemble generation process, while a more robust loss function or hybrid methods that leverage strengths of numerical techniques could increase performance and reliability.

### 5 Conclusions

Our resource-efficient, data-driven methodology demonstrates skillful results for the estimation of seasonal surface chl-a at a near-global scale. This is done by using MLD, SSH, SSS and SST data as input for a lightweight, U-Net-like neural network that is trainable in a matter of hours on a single GPU. Our method can be used to reconstruct historical chl-a patterns (i.e. by using a reanalysis as input), or to generate predictions based on easily accessible forecast data. These predictions maintain low error and high correlation to observations across different regions and lead-times. The method is computationally efficient, and can be easily expanded to improve accuracy and extend its capabilities.





*Author contributions.* J.J, R.B and G.M.B conceived the study. G.M.B collected the data, developed the model, performed the data analysis and created the figures and original draft of the manuscript. All authors discussed the results and contributed to the final manuscript.

*Competing interests.* SC is a member of the editorial board of Biogeosciences.

*Acknowledgements.* This work was funded by the project DTO-Bioflow, which has received funding from the European Union's Horizon Europe Innovation Action under grant agreement No 101112823. Part of the work was supported by the project NECCTON ("New Copernicus Capability for Trophic Ocean Networks"), which has received funding from Horizon Europe RIA under Grant Number 101081273.



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
