# Peer review of "Estimating Seasonal Global Sea Surface Chlorophyll-a with Resource-Efficient Neural Networks"

_EGUsphere, 2025_

## Author Comment (AC1)

**Comments from Reviewer 2:**

*This article presents an interesting data-driven approach to predict chlorophyll-a (chl-a) fields on a near-global scale based on seasonal forecasts of some oceanic physical properties, which offers an alternative tool to mechanistic biogeochemical models. Previous efforts in data-driven global chl-a reconstruction from oceanic physical properties have primarily targeted long-term retrospective analyses. In contrast, this study explores shorter-term seasonal forecasts of chl-a and argues that the results compare favorably with biogeochemical models. The topic is timely, the scientific question is original and clearly formulated, and the overall structure of the manuscript is easy to follow. However, I think several points would require further elaboration and justification to make the manuscript more rigorous and convincing.* We thank the reviewer for their positive comments and appreciate their constructive feedback. We have addressed their comments in detail below.

**General comments**

*1) Choice of predictors. The rationale for selecting only four predictors deserves further justification. While they are relevant, some may contain redundancy and some processes are not explicitly considered. It would help to discuss assumptions about neglected drivers and how this may influence the spatial and temporal variability of results. For instance, light availability is a key driver, particularly at high latitudes and in some tropical regions (see for instance Fig3A-B of Racault et al., 2017). Although SST may correlate with PAR seasonally, this relationship does not hold consistently at inter-annual timescales. Why was this predictor not included, for example? Is the potential improvement in performance considered negligible compared to the gain in terms of computing time for model training?* We thank the reviewer for pointing this out. We agree that light availability is a key driver of chlorophyll-a variability. However, PAR was not included in this study because it is not available in the SEAS5 forecast product nor the GLORYS12 reanalysis, making it difficult to integrate into our current forecasting pipeline. We will note this limitation in the Discussion. Incorporating this driver is a priority and we are actively exploring ways to include it in future work.

*2) Temporal resolution. The use of 5-day data for training, while the final application relies on monthly inputs, is not fully explained. Clarifying the advantages and limitations of this choice would help readers understand whether it may contribute to the underestimation of inter-annual variability.* We thank the reviewer for highlighting this; we realize that this was not properly presented in the manuscript. We developed two versions of the model: one trained on monthly data to align with the publicly available SEAS5 forecasts, and another on 5-day averages to test higher temporal resolution. While higher resolution may improve sensitivity to anomalies, we focused on monthly data to capture large-scale trends and match SEAS5's availability. We will clarify this rationale more clearly in the manuscript.

*3) Reproducibility and robustness. Certain technical details are not sufficiently specified to ensure the reproducibility of the experiments (e.g., learning rate, number of training epochs, criteria applied to stop training, etc) and to convince readers of the robustness of the approach.*

*While possible overfitting is mentioned to explain the network's weaker ability to predict inter-annual variability than seasonal one, giving more details regarding any regularization techniques used to monitor and limit this would strengthen confidence in the robustness of the approach.* We thank the reviewer for highlighting this, we realize that this was not clearly articulated in our methodology. We will entirely revise the methodology section to make it more specific and detailed.

*4) Seasonal vs inter-annual variability. The manuscript could be strengthened by further analyses and discussion on how the data-driven approach performs relative to mechanistic models in representing inter-annual variability. This variability is more difficult to reproduce but highly relevant for societal applications (e.g., ENSO impacts), whereas reproducing the climatological seasonal cycle is comparatively less challenging.* We appreciate the reviewer's comment, we agree on the importance of inter-annual skill, particularly in the context of societal applications. We include results from four distinct years (2020–2023) to incorporate a range of inter-annual conditions, but we acknowledge that this is a relatively short period. We will revise the manuscript to discuss this limitation. A quantitative comparison to mechanistic model forecast skill for the same period is challenging since we don't have access to these (and BIO4 forecasts are limited to 10 days), but we will consider including a comparison to the analysis for this timeframe.

*5) Machine learning positioning. Clarifying the distinction between the efficiency of the overall data-driven framework versus the neural network architecture itself would avoid confusion. Six million parameters can be seen relatively large compared with some published CNNs, and the choice of U-Net over simpler alternatives could be justified more explicitly in terms of added value.* We will revise the manuscript to clarify this distinction and elaborate on the choice of the U-Net.

*Specific comments*

*Title : I would have rephrased the title into something like : « Forecasting Seasonal Global Sea Surface Chlorophyll-a with a lightweight data-driven approach » to emphasize the forecasting dimension and the efficiency of the overall method (vs. the efficiency of the architecture itself).* We thank the reviewer for the title suggestion. We agree that their suggestion captures key aspects of the manuscript, and we will take it into consideration as we finalize the revised version.

*L4 : I would recommend a sentence like : « We propose a data-driven resource-efficient alternative : a neural architecture based on the U-Net that reconstructs surface, [..] from four physical predictors »; L15 : I think « long-term » is not appropriate here, I would recommend « seasonal »* We will incorporate these suggestions into the manuscript.

*L69-70 : « While advances in computational power and data availability have enabled the development of more complex architectures, the simplicity and efficiency of the U-Net makes it an effective and resource-efficient choice for this task ». Although the U-Net architecture may appear simpler than more recent transformer-based models, it is still more complex than basic CNNs with fewer parameters that have been used in some previous studies. While I am*

*convinced that a Unet is well suited for this application, I would have justified this choice differently, for example by emphasizing its ability to better capture different spatial scales.* We thank the reviewer for their comment. We will revise the manuscript to elaborate on the choice of the U-Net.

*L77 : « The hyperparameters [...] were optimized using random search ». Please provide a detailed description of all hyperparameters used and clarify on which dataset they were tuned: was optimization based on the training period (1998–2017) or the validation period (2017–2023)? If the latter, how did the authors account for potential overfitting? Why was an entirely independent time period not used to assess the model's generalization performance more objectively?* We appreciate the reviewer's suggestion. We realize that the divide was not clearly articulated in our methodology. We will clarify this more thoroughly in the revised manuscript and include more detail with regards to the hyperparameters used. *Could the authors also provide the training and validation loss curves? Finally, did the authors check that the model's learning was stable from one run to the next?* While we did monitor the training and validation loss curves during the training process, we did not save these and would need to retrain the models to generate them. Learning stability was consistently observed and we used early stopping to prevent overfitting; therefore, we believe that including these curves is not essential for the current manuscript.

*L85 : « To simplify the predictive task, which consists of using a 6-month forecast of the physics to predict chl-a for the same time period, we trained twelve dedicated neural networks, each corresponding to the starting month of the forecast ». The choice seems to increase methodological complexity while reducing the training data available for each model, potentially favoring overfitting. I would have clarified how this risk was assessed and justified this strategy more explicitly, including its potential advantages and drawbacks. Have the authors compared reconstruction performance over the six-month period when using a single model trained across the whole dataset versus the proposed approach based on twelve separate models?* The reviewer raises a valid concern. This approach was chosen because it resulted in smaller, more lightweight models compared to a single network that would require more parameters to simultaneously handle all variability. It also aligned with the practical operational application that we had in mind, where users generate chl-a predictions starting from a month determined by the available physics forecast. However, we acknowledge that this choice may limit the models' ability to generalize across different initializations and potentially reduce the model's ability to learn universal temporal relationships. We will revise the manuscript to make this clear.

*L90-92 : « For context, [...], has between 20 and 30 million ». I would have removed that sentence. In Ansari et al. (2022), some of the Unets mentioned have almost six times fewer parameters. If a comparison in terms of parameters is to be made, I think it would be more relevant to compare them with other deep learning models published on the topic of Chl reconstruction.* We appreciate the reviewer's comment. Our intention was not to benchmark against other lightweight models or existing Chl-a studies (especially as we are not aware of reported parameter counts for the latter). We simply wanted to highlight that, despite the 3D convolutions, our network in fact has fewer parameters than the original 2D U-Net, which we

found to be interesting and somewhat unexpected. We will revise the manuscript to make this motivation clearer.

*L104 : « Since remotely sensed chl-a is limited by sunlight availability, we focus on the -60° to 80° latitudes ». Spatial coverage is not symmetric in latitude, suggesting factors beyond sunlight, e.g., cloud-dependent pixel availability (higher in the Southern Ocean). I would have justified the footprint selection more objectively, for example using a minimum pixel density.* We thank the reviewer for this observation. While we acknowledge that the use of minimum pixel density could provide a more objective criterion, we have chosen to focus on the latitude range to simplify our analysis as our data are on a regular grid. Since GlobColour is a merged product that is already gap-filled, our major constraint was linked to sunlight availability, which limits coverage at high latitudes. We will revise the manuscript to clarify this point.

*L117-L119 : « The SEAS5 forecast [..]. To avoid these from biasing […], and then subtracting this difference from SEAS5 before using it as input ». I am not sure I understand the rationale of this approach. When applying the model to future forecasts, the corresponding GLORYS12 reanalyses will not be available—how will this be addressed? Is only a fixed annual climatology subtracted?* We thank the reviewer for this observation. We correct SEAS5 with respect to the *historical* climatologies of both datasets. We will clarify this point in the manuscript. We also note that similar results were obtained when using the uncorrected SEAS5 forecast (we will include the figures in the Supplementary Materials); the bias correction was applied primarily to isolate model performance evaluation.

*L165 : « the ACC decreases with lead time due to the increased uncertainty in the forecasted physics». For Figure 6b (6-month lead-time ACC), have the authors checked whether the poorly predicted (blue) areas correspond to regions where the seasonal forecasts of the four physical predictors are less accurate, based on literature or error maps?* We appreciate the reviewer's suggestion. We have not yet conducted a detailed comparison between the poorly predicted areas in Figure 6b and forecast skill for the physical predictors. We acknowledge that this would provide valuable context and will consider including such an analysis in future work. We will also note this limitation in the manuscript.

*In the discussion (L235-237), the use of biogeochemical variables as predictors is mentioned. Can't these forecasts carry larger errors than the physical fields, and couldn't their inclusion risk degrading six-month forecast quality?* The reviewer raises a valid concern. In this study, we chose to focus exclusively on physical predictors to leverage the reliability of reanalysis data and indeed to avoid potential uncertainties associated with biogeochemical variables. As with any modeling approach, trade-offs are inevitable. While biogeochemical variables can carry higher uncertainties, they may also capture important feedbacks not fully represented by physical predictors. We will clarify this rationale in the manuscript.

*FIG 7 : Have the authors tried plotting seasonal chl-a forecasts initialized for several months using SEAS5, in the same way as Figure 5, across the different regions? Providing these outputs in the Supplementary Material could help assess spatio-temporal heterogeneity in model performance over multiple lead-time months.* We thank the reviewer for this helpful suggestion. We agree that presenting forecasts initialized for multiple months across different

regions would provide valuable insight into the spatio-temporal heterogeneity of model performance. We will generate these plots and include them to support the evaluation.

*L169-170 : « These figures demonstrate that the neural network is able to capture the seasonal dynamics of the data across regions, regardless of the input physics. » This statement could be complemented by a discussion on interannual variations. ACC metrics for the different regions could support this discussion.* We thank the reviewer for their suggestion. We agree that complementing the current discussion with an assessment of interannual variations could add value, we will revise the manuscript to include this. Incorporating regional ACC curves could also provide additional insight and we will consider including them as part of our extended analysis.

*FIG8 & 9: Figure 8 is currently under-described; a more detailed analysis is recommended to strengthen the argument. For Figure 9, adding ACC curves would provide a metric specific to interannual variations and show their evolution over time.* We agree that a more detailed analysis of Figure 8 could further strengthen our argument. We will enhance its description in the revised manuscript.

*L190 : « a simple neural network » : I would remove the term « simple ». Similarly, L217, I would either remove « shallow », or insist on the fact that this is a lighter approach than a classical mechanistic model. L246 : I would recommend removing « lightweight »* We thank the reviewer for this observation. Our use of terms such as "simple," "shallow," and "lightweight" was intended to emphasize the relatively low complexity and computational cost of our model compared to current state-of-the-art AI forecasting systems. We will revise the manuscript accordingly to convey this more clearly.

*L249 : « These predictions maintain low error and high correlation to observations across different regions and lead-times ». I would qualify that statement, as this may not be the case with regard to interannual variability.* We agree that interannual variability can affect predictive performance. In the manuscript, we report results for four different years (2020-2023, included) to capture some of this variability. However, we acknowledge that this represents a limited sample. We will revisit this statement to better reflect this limitation.

*Technical corrections* We thank the reviewer for these observations, and in particular for highlighting the issue with the lead times. We mistakenly labeled lead-time 6 as lead-time 5 on figure 2 and will correct this for the revised manuscript.

We will correct this, along with all the other edits suggested below, in the revised manuscript.

*L26: even 'when' limited instead of « even if limited » ?*

*L44 : « variables » or « predictors » instead of « parameters »*

*L60 : « available forecast of these as input » may be replaced by « available forecasts of those former variables as input» for clarity ?*

*L83 : « the physical ocean data was » → the physical ocean data were*

*L91 : « Ronneberger et al. »→ please precise the date of the reference.*

*L94-97 : Please precise the initial spatio-temporal resolution of the different used dataset, as well as the final resolution used in this study. If pre-processing was made (resampling, etc), please precise it.*

*L110 : « it uses the IFS cycle 43r1 for the atmosphere, NEM0 for the ocean, and LIM2 for sea ice ». Please precise the acronym and what it corresponds to for non numerical modeling readers.*

*Fg2 : Perhaps I have misunderstood, but why not have six reconstructed time series, with lead times 1-6 (see lead time of 6 shown in Figure 6) instead of five in this diagram ?*

*Fig 3 : it would be great to have the lat/lon axis plotted on the maps.*

*FIG 7 : I would recommend adding on the Figure some quantitative correlation metrics between the different time series to better compare performance between areas.*

**References**

*Racault, M. F., Sathyendranath, S., Brewin, R. J., Raitsos, D. E., Jackson, T., & Platt, T. (2017). Impact of El Niño variability on oceanic phytoplankton. Frontiers in Marine Science, 4,133*

---

## Author Comment (AC2)

**Comments from Reviewer 1:**

*The authors use a neural network to estimate surface chlorophyll-a, a computationally efficient approach that appears to outperform traditional approaches like mechanistic biogeochemical ocean models. The manuscript presents some compelling results, but the experimental setup is not described well enough, and it is unclear why the comparison of chl-a estimates does not include any coastal regions.* We thank the reviewer for their positive comments and appreciate their constructive feedback. We have addressed their comments in detail below.

**general comments:**

*The manuscript is mostly well written and was easy to follow -- with a major exception: the basic setup of the experiments and implementation details are not well described and after reading through the whole manuscript I still do not quite know what, for example, "6-month predictions" are in the manuscript. Does the "6-month" imply a 6-month lead time, a 6-month forecast length, a 6-month time average or something else? Is there a distinction between "prediction" and "forecast" in the manuscript, if so, what is it? Sentences that are meant to explain experiments sometime increase the confusion of the reader, for instance: "These months correspond to lead-times one out of the six months of each forecast." (l. 156). Sentences like this example are confusing to the reader and could be improved considerably by rephrasing and adding some details. Please take the time and space to clarify how the experiments are set up and what is compared at what resolution (this includes space and time).* We thank the reviewer for highlighting this, we realize that this was not clearly articulated in our methodology. By "6-month predictions," we refer to forecasts with a 6-month horizon (i.e., predicting conditions up to 6 months ahead). We also evaluate performance at different lead times by analyzing forecast skill separately for each month within that horizon (i.e. lead times 1-6 *months*). This will be clarified in the revised manuscript.

*Even a reader who does not know much about marine chl-a might find it surprising that the regions where performance evaluated, shown in Fig. 3, do not include any "yellow" values and seem to focus only on open-ocean regions (as an aside, a color bar or at least a description of what property is shown in Fig. 3 would be useful). That is, why weren't any coastal regions with high chl-a concentrations included in the comparison? The authors mention "fisheries management" and "harmful algal blooms" but then neglect to evaluate the model in the biologically active regions where most blooms occur and fishery is prevalent. In general, the chl-a estimates were compared mostly as a global average (Fig. 4, 5) or as averages in the large open-ocean regions (Fig. 7, 9); only Fig. 6 shows the performance on a finer spatial scale.* We thank the reviewer for this insightful comment. We agree that the current regional evaluation, which focuses on open-ocean areas, does not fully represent the biologically active coastal regions that are relevant to the applications mentioned in the introduction. We are currently considering how best to address this limitation, either through a more detailed analysis of these regions, a revision of the discussion and introduction to better reflect the current scope, or both.

Additionally, we will improve the clarity of Figure 3 by adding a color bar and updating the figure caption.

*Even in the computation of the RMSE, a spatial average appears to be used: "The spatially-averaged reconstructed time series has a RMSE of 0.01 ..." (l. 151). Why is the RMSE based on a spatial average? The use of spatial averaging is not explained well or mentioned when the RMSE is introduced. Please ensure that the reader knows at all times how key metrics are being computed.* We thank the reviewer for highlighting this. We agree this aspect of the Methodology section was unclear and will revise the manuscript accordingly. We will ensure that the definitions for all key metrics are clearly described to avoid ambiguity.

*In addition, I would suggest using nearshore regions in the comparison and evaluating the model performance at a higher resolution, both in space and time.* We agree that evaluating performance in nearshore regions could offer valuable insights. While higher spatial or temporal resolution is not possible with the current model configuration (because resolution is fixed), we are exploring the feasibility of assessing model skill in nearshore areas using the existing output. We recognize the importance of this direction and consider it a key area for future development.

*Furthermore, the authors later ponder how the decrease in ACC observed in Fig. 6 aligns with little to no increase in RMSE and other metrics in Fig. 9. They explain that "it is likely that the neural network's ability to capture the strong seasonal dynamics in the data (Figs. 7 and 8) is compensating for the decrease in performance with respect to the anomalies" (l. 168). That could well be, but if the RMSE is based on some spatially averaged chl-a, the averaging could have removed most of the effect of the anomalies. Unfortunately, a reader can only guess here, as it is unclear how the RMSE was computed.* We thank the reviewer for pointing this out. The RMSE reported in that section is calculated over all spatial points and time steps, but we recognize that the description could be clearer. We will revise the manuscript to explicitly clarify this calculation and its implications.

*Due to their distribution, when plotting and comparing chl-a values, they are often log-transformed. The authors mention once that a log-transformation was used, but it is unclear where and to what extent: "The physical ocean data was normalized using min-max normalization and the chl-a data was log-transformed" (l. 82) is the only information the reader gets. Was a log-transformation used when computing the ACC, NRMSE etc., are $r_i$ and $p_i$ in Eq 1-4 log-transformed? How were the climatologies computed? More importantly, perhaps, was a log-transformation used in the loss function for the neural network? The authors mention that they needed to modify the loss function: "so we modified the standard mean squared error (MSE) loss function by adding a small penalty for underestimation." (l. 79). With a log-transformation applied to chl-a, one would expect underestimation to be quite heavily penalized by the MSE. More information is needed to better interpret the results and the setup of the neural network.* We thank the reviewer for these important questions and recognize that these details were not clearly described in the original manuscript. To clarify, the evaluation metrics (ACC, NRMSE, etc.) are computed on the original scale, once the neural network output has been transformed back to the natural scale. However, the network predicts log-transformed chlorophyll-a values, and the loss function is calculated on log-transformed. Despite this

approach, we observed underestimation during training, motivating our modification of the standard MSE loss. We will revise the manuscript and update the notation in the equations to clarify these points.

*specific comments:*

*L 1: "Marine chlorophyll-a is an important indicator of ecosystem health, and accurate forecasting, even at the surface level, can have significant implications for climate studies and resource management a lightweight, resource-efficient neural architecture based on the U-Net that reconstructs surface, near-global chlorophyll-a from four physical predictors.": Accurately forecasting/estimating surface chl-a is a good check for "traditional" mechanistic models to verify that they can recreate some key biogeochemical dynamics. How would the output of a neural network model that only estimates surface chl-a be able to inform climate studies and resource management? Maybe this is a point that could be discussed further in Section 4.* We thank the reviewer for raising this point. While our model focuses solely on surface chlorophyll-a, this variable can be used as a proxy for phytoplankton biomass and primary productivity and has the advantage of being easily verifiable through observations. Surface chl-a estimates can provide insights into ecosystem health, carbon cycling, and responses to climate variability. However, we acknowledge the importance of subsurface processes and vertical structures, which are not captured by our approach. We will expand the discussion in Section 4 to address these strengths and limitations.

*L 59: "The goal of this work is to demonstrate that we can not only estimate chl-a from these four variables, but that by using publicly available forecasts of these as input, we are able to generate an ensemble of skillful chl-a predictions for six months into the future.": Here it would be useful for the reader to be more specific: are the 6-month predictions reliant on a 6-month forecast or are they produced from input 6 months into the past?* We thank the reviewer for this question. To clarify, the six-month chlorophyll-a predictions are directly reliant on the six-month forecasts of the physical variables used as inputs. We will revise the manuscript to explicitly state this for clarity.

*L 74: "Skip connections link matching layers in the encoder and decoder, facilitating the transfer of information.": Does this mean the first Conv3D layer is linked to the last one, etc.?* Not exactly; the fourth convolutional layer in the encoder is concatenated with the output of the first upsampling operation in the decoder, and the second convolutional layer is concatenated with the output of the second upsampling layer. We will revise the Methodology section and Table 1 to clarify this.

*Eq 1: It would be good to explain the terms in the equation a bit better (is the data log-transformed?) and move the equation up to where MSE and the terms are introduced.* We thank the reviewer for highlighting this point; we recognize that the description could be clearer. We will revise the manuscript to explicitly clarify all metric calculations.

*L 85: What motivated the choice of the 12 "monthly" neural networks? How much worse is the use of a single one for all months?* This approach was chosen because aligned with the practical operational application that we had in mind, where the user would generate the chl-a

prediction starting at a given month that depended on the physics forecast available. We considered that this would result in smaller, more lightweight models that would be used according to the initialization month of interest. However, we acknowledge that this design choice may limit, by design, the models' ability to generalize across universal temporal relationships. We will revise the manuscript to include this.

*L 90: "The optimal architecture found for this task has approximately six million trainable parameters...": Is this for one or all 12 of the networks?* We thank the reviewer for this question; we realize that we did not provide sufficient detail on this point in the manuscript. Each one of the 12 networks has approximately six million parameters, and although we employ transfer learning by initializing one network using the weights from others, all parameters remain trainable and independently updated. We are aware that this is not the most efficient approach, as transfer learning could be further leveraged through parameter sharing or freezing to reduce the total count. We will clarify this in the manuscript.

*L 97: "...provides daily and monthly data...": Here, or somewhere early on, mention if the networks produce daily or monthly mean estimates.* We thank the reviewer for highlighting this point; we recognize that the description could be clearer. We will revise the manuscript accordingly.

*L 122: "lead-time two": Does this mean a 2-month lead time?* Yes. We will revise the manuscript to clarify this point.

*Eq 2-4: How do these metrics compare to the cost function used for training the network, why not report/show that value as well? And mention if any of these chl-a values are log-transformed in these metrics.* We thank the reviewer for this comment. The cost function used during training is based on the log-transformed chlorophyll-a values, but the evaluation metrics reported in the manuscript are computed on the original (non-log-transformed) scale. We will revise the manuscript to clarify this distinction. Regarding the training loss, we chose not to report it, as it is not commonly included in the evaluation of machine learning models where the focus is typically on independent performance on data that the model has never seen before.

*L 146: "Rather than a direct comparison, we use BIO4 as a benchmark, recognizing that it simulates a wide range of interconnected biogeochemical processes across various depths, whereas our data-driven approach is specifically designed for surface chl-a prediction.": This sentence is a bit confusing. It makes sense to compare the neural network approach to a more classic reference approach for estimating surface chl-a. But why is this dependent on BIO4 also estimating a wide range of other properties? Maybe I just do not understand what "direct comparison" refers to in this context.* We thank the reviewer for the comment. Our intention was to note that while we compare surface chlorophyll-a predictions, BIO4 is a more complex model that also simulates a broader range of biogeochemical processes. We will revise the sentence to clarify that BIO4's broader scope is mentioned for context.

*L 150: The first sentence of Sec 3 is almost identical to that of Sec 2.2. Unfortunately, it is still not clear to me what a "set of 5-day predictions" means.* We thank the reviewer for pointing this out. We will revise both sentences to improve clarity.

*L 151: "The spatially-averaged reconstructed time series...": What kind of spatial averaging is performed here, before computing the RMSE etc.?* All spatial points are averaged to obtain a single time series, and the metrics are then calculated on this averaged series. However, metric values computed without spatial averaging are reported elsewhere in the manuscript, we will clarify this distinction to avoid confusion.

*L 168 and following figures: Are the BIO4 estimates that are shown forecasts as well? For what lead time?* No, the BIO4 estimates are not forecasts, they are outputs from the analysis itself. We will clarify this in the manuscript.

*Citation: https://doi.org/10.5194/egusphere-2025-1246-RC1*